# Study of Chemotherapy-Induced Cognitive Impairment in Women with Breast Cancer

**DOI:** 10.3390/ijerph17238896

**Published:** 2020-11-30

**Authors:** Blanca Rodríguez Martín, Eduardo José Fernández Rodríguez, María Isabel Rihuete Galve, Juan Jesús Cruz Hernández

**Affiliations:** 1Medical Oncology Service, Salamanca University Hospital, 37007 Salamanca, Spain; edujfr@usal.es (E.J.F.R.); rihuete@usal.es (M.I.R.G.); jjcruz@usal.es (J.J.C.H.); 2Nursing and Physiotherapy Department, University of Salamanca, 37007 Salamanca, Spain; 3Medicine Department, University of Salamanca, 37007 Salamanca, Spain

**Keywords:** chemotherapy, cognitive impairment, cancer

## Abstract

Background: Oncology patients experience a large number of symptoms and, those referring to cognitive performance has an ever-increasing importance in clinical practice, due to the increase in survival rates and interest in the patient’s quality of life. The studies reviewed showed that chemotherapy-related cognitive impairment might occur in 15 and 50% of oncology patients. The main objective of this research was to study the impact of chemotherapy on the cognitive function of patients with locoregional breast cancer. Method: Analytical, prospective, longitudinal study using three measures, unifactorial intrasubject design, non-probability, and random selection sampling. The sample comprised women newly diagnosed with locoregional breast cancer in stages I, II, IIIA who received chemotherapy at the University Hospital of Salamanca (Complejo Asistencial Universitario de Salamanca), randomly selected for three years. Semi-structured interviews were conducted, and anxiety and depression (Hospital Anxiety and Depression scale, HAD); quality of life (QLQ-BR23 scale) and the following cognitive variables were assessed—processing speed, attention, memory, and executive functions (subtests of the Wechsler Intelligence Scale and the Trail Making Test). Results: The final sample size included 151 participants; 23 were excluded. A decline in cognitive performance was observed in patients, which did not completely recover two months after chemotherapy was completed. Additionally, worse cognitive performance was observed in patients with anxious or depressive symptoms. There was a negative impact on the quality of life. Conclusion: Chemotherapy had an impact on the cognitive performance of oncology patients in most cognitive domains studied.

## 1. Introduction

The quality of life of oncology patients is one of the aspects of most concern for healthcare professionals in their clinical practice, especially in the last two decades. This increased interest in the impact of oncological treatments on the cognitive functioning of patients.

Most research on cognitive functioning was performed on women with breast cancer, as it is a large population, their cognitive functions are in better conditions, and they have fewer symptoms than patients with other oncological pathologies. Additionally, the high survival rates of these patients show the importance of cognitive disorders such as memory loss or attention and learning difficulties. These symptoms can go unnoticed in the pathology of other types of cancer, such as lung cancer, head and neck tumours, or gastric cancer, since there are other priority symptoms.

The first studies that were performed did not have well-defined designs, and their small sample size did not allow conclusive outcomes. Most studies were cross-sectional, and the evaluation of cognitive performance was carried out at a single moment in time, during or after chemotherapy [1]. This variability among measurements made it difficult to compare results.

Most studies include the requirement of assessing pre-treatment baseline cognitive performance to compare future results. These first evaluation studies accurately predicted the extent of the change observed after oncological treatment, and at the same time, its development over time. The importance of the first evaluation was evidenced in the study by Wefel [1], a longitudinal study that includes a pre-treatment evaluation and concludes that cognitive impairment might be present before treatment. Results indicate that 35% of the sample presents cognitive impairment. Verbal learning (18%) and memory (25%) are the most affected domains.

Other studies such as Hurria and Hermelink [2,3] also showed the presence of basal cognitive impairment. In the first study, 11% of the sample presented cognitive impairment in the baseline and in the second study, the group’s average had a lower performance compared to the data published in 5 of the 12 tests administered to evaluate cognitive functioning. 

This baseline impairment could be explained by the presence of personal or environmental variables, such as emotional state, the hemoglobin level or a low educational level.

It is not only necessary to know the baseline cognitive status of the patients before the administration of chemotherapy to have a comparative reference, but it would also be substantial to measure at different times of the treatment, to identify the evolution in the cognitive performance of the patients, since previous studies found that cognitive impairment increases after chemotherapy [4]. Given the results of the different studies, we observe that cognitive impairment in oncology patients undergoing chemotherapy is a factor to be taken into account, given its prevalence (between 15–35%) in the overall assessment of patients [1].

The mechanisms that cause chemotherapy-induced cognitive impairment in cancer patients receiving chemotherapy are not yet well-known, although there exists multiple etiology.

The factors that the literature proposes as most influential in chemotherapy-induced cognitive impairment are
-Direct neurotoxic effects—cytostatic drugs that cross the blood–brain barrier can cause cell death [5].-Induced hormonal changes—these changes can interfere with hormone secretion and activate cognitive problems. It is known that chemotherapy changes the testosterone and estrogen levels, which are considered neuroprotective hormones.-Oxidative stress—chemotherapy decreases the cellular antioxidant capacity and thus increases DNA damage [6].-Immune system dysregulation caused by cytokine release—inflammatory cytokines cross the blood–brain barrier and can cause a decline in cognitive function, manifested as decreased processing speed, executive function, spatial ability, and reaction time [7].-Vascular damage—coagulation in small vessels of the central nervous system, vascular damage, and autoimmune phenomena. Both chemotherapy and radiotherapy can damage blood vessels, which reduce blood flow in the small blood vessels in the brain [8].

### 1.1. Causes that Speed up Impairment

Cognitive impairment is the loss of cognitive functions (memory, attention, and speed of information processing) that occur in normal ageing. 

Cognitive domains function correctly when the brain structure and function is in an optimal condition. Chemotherapy crosses the blood–brain barrier, alters mental functioning, and causes impairment of some cognitive domains. 

The cognitive impairment of our brain depends on both physiological and environmental factors and is subject to significant interindividual variability [9].

Hess [10] also proposed direct and indirect treatment effects on cognition. However, chemotherapy does not act as an isolated entity, but the size of the effect is controlled by factors that increase or reduce the former vulnerability. 

Factors that might speed up the cognitive impairment process are:
-Chemotherapy-related cognitive impairment—toxicity might produce cognitive changes when crossing the blood–brain barrier.-Stress-related cognitive impairment—stress can negatively affect memory as it increases cortisol released by the adrenal glands, and this substance directly affects the hippocampus, which is part of the limbic system dedicated to working and short-term memory [11].-Anxiety-related cognitive impairment—excessive worry and irrational fear can impair memory by focusing thought on a particular obsession. Memory suffers from anxiety and can cause memory loss [12].-Depression-related cognitive impairment—depression might be related to attentional problems, which affect the information acquisition and coding phase. The data provided by different scientific studies show adverse neuropsychological effects of chemotherapy [13].

The meta-analyses focused on this subject have selected some of these studies [14,15,16]. Some authors made an effort to correct the effect of longitudinal analysis [17] to show the existent degree of impairment. Others like Heflin et al. [18] made an effort to control genetic and shared environmental variables to assure that only cancer and treatment variables are the cause of cognitive impairment in some patients. Although caution is required to interpret the findings due to the methodological limitations of the studies performed, all studies suggest the existence of cognitive impairment, supported by the test results, as the causes for such damage are still uncertain. 

A dose-dependent correlation was found [19,20] between the dose and the time elapsed since the last chemotherapy administration, which are important factors in the remission of the chemotherapy-induced deficits. There exists a negative correlation in these variables (the longer the period since the last cycle of chemotherapy, the better the neuropsychological test performance). 

The most positive results assure that after six months, the impairment disappears entirely, or there tends to be a progressive recovery [21]. 

This suggests that the cognitive impairment of the affected patient is more likely to be transitory in patients treated with fewer cycles of chemotherapy, at lower doses, over a long period. Patients who recently finished their treatment, with a larger number of cycles and high doses of the drug in each cycle, showed significant impairment. The bibliographic review shows that 15 to 50% of oncology patients receiving chemotherapy present some type of neurological complication with regards to cognitive functioning in one or more domains [22,23].

The studies consulted define some cognitive impairment in oncology patients before the treatment and suggest the possibility of an increase in the impairment, during therapy, as a consequence. It is convenient to explore its influence on cognitive functioning two months after starting the treatment, and also study if the possible impairment is maintained for two to six months after its completion [16,24,25].

Patients with breast cancer are more susceptible to cognitive impairment, due to their treatments and because their tumour causes fewer symptoms than other cancers, so they are likely to pay more attention to cognitive issues.

A longitudinal study was performed that allowed us to verify the pre-treatment cognitive impairment of oncology patients, and increase in impairment post-treatment and after the treatment was completed.

### 1.2. Study Hypotheses and Objectives

Based on the above, the following hypothesis was stated: 

Patients with locoregional breast cancer present throughout their treatment a cognitive impairment that was preserved at least two months after its completion.

The main objective was to study the influence of chemotherapy on the cognitive status of patients with locoregional breast cancer at the University Hospital of Salamanca (Complejo Asistencial Universitario de Salamanca (CAUSA from its Spanish initials)).

Specific objectives:To analyze the impact of chemotherapy on the cognitive domains in the three study periods measured.To assess whether the emotional state affects the cognitive performance of patients.To define if there are changes in the quality of life of the patients during the treatment and if this affects their cognitive performance.

## 2. Materials and Methods 

### 2.1. Study Design

Analytical, prospective, longitudinal study using three measures, unifactorial intrasubject design, non-probability, and random selection sampling. The sample was provided by the Medical Oncology Service of the University Hospital of Salamanca.

### 2.2. Participants

The scope of the study comprised patients newly diagnosed with locoregional breast cancer in stages I, II, and IIIA, and who are about to receive chemotherapy at the Day Hospital and the Medical Oncology Hospitalisation Unit.

All patients were newly diagnosed and received adjuvant or neoadjuvant chemotherapy, and did not receive any previous oncological treatment.

The patients were randomly assigned to the study between 2015 and 2017.

The following inclusion criteria were established—female; anatomopathological diagnosis of locoregional breast cancer; fit to receive chemotherapy; is a patient who attends the University Hospital of Salamanca; is of legal age and less than 85 years-old; stage I, II, and IIIA breast cancer; and written informed consent stating that participation was voluntary. Additionally, a series of exclusion and withdrawal criteria were stated, as shown in Table 1.

### 2.3. Procedure

A research project and a viability check by the Salamanca University Assistance Complex’s ethics committee to implement this research was developed. 

The recruitment of patients started in October 2015, when the research was approved. A baseline was defined at the beginning of the chemotherapy.

Those patients who verbally consented to participate in the study were called to the Medical Oncology offices to sign the written informed consent and to explain the objective and procedure of the study.

All assessment was performed on the same day the patients received the chemotherapy, between the analysis and the beginning of the chemotherapy cycle.

In the first interview, clinical and socio-demographic data were collected from the patients, and the baseline was set using the tests indicated above.

The evolution of the patients was followed up during the three months of chemotherapy treatment and within two months of the end of treatment.

Therefore, three measurements were established to study the evolution of cognitive performance—at the beginning of chemotherapy, three months after receiving treatment, and finally, two months after the treatment ended.

These three measurement moments were established with the aim of evaluating the effect chemotherapy had on the cognitive performance of the patients studied. A total of 613 interventions of approximately one hour were performed.

This set of tests was suitable for application in this type of research, specifically in people suffering from an oncological disease, for its selection by relevant psychometric issues, the high incidence of use in similar studies, and its brief and dynamic application. 

### 2.4. Description of the Variables

The variables anxiety, depression, and quality of life were studied, as well as their cognitive function, analyzing processing speed, attention, memory, and cognitive functions.

### 2.5. Evaluation Tools

Anxiety and depression were evaluated by the Hospital Anxiety and Depression Scale (HAD) [26], quality of life evaluated by the European Organisation for Research and Treatment of Cancer Quality of Life Questionnaire C-30 version 3 (EORTC QLQ-BR23 Scale) [27]. Cognitive variables: processing speed, attention, memory and executive functions were evaluated by the subtests of the Wechsler Intelligence Scale [28] and the Trail Making Test (TMT) [29]:-Processing speed—measured by the Symbol Search and Key Search Subtests.-Attention—measured by the Trail Making Test and the Stroop Color and Word Test [30].-Memory—measured by the Vocabulary Subtest.

These evaluation tests were considered due to their high use in cancer patients, and also because they are standard tests of choice in the study of cognitive and attentional functions and health-related quality of life.

In addition, a semi-structured interview was conducted with all patients for the collection of clinical and socio-demographic data.

Data collection took place from October 2015 to January 2017.

The study was conducted after the authorization of the Clinical Research Ethics Committee of the Salamanca Health Area, and prior informed consent of the study subjects, following the Declaration of Helsinki. The participants were informed of the objectives and the risks and benefits of the studies (informed consent). Similarly, the confidentiality of the subjects included was guaranteed at all times, following the provisions of the Organic Law 3/2018 of 5 December on the Protection of Personal Data and Guarantee of Digital Rights and the Regulation (EU) 2016/679 of the European Parliament and Council of 27 April 2016, on data protection (GDPR), and the conditions established by Law 14/2007 on biomedical research. Approval number—0000263. Name of the Council—Bioethics Committee of the University of Salamanca. Board Affiliation—University of Salamanca.

### 2.6. Statistical Analysis

The statistical analysis was pre-planned, with small modifications once the study was performed. Initially, an exhaustive review and data cleansing was carried out to detect possible errors in the data collection, and to correctly and accurately apply the exclusion criteria established for the study. For this purpose, a descriptive analysis was used, focusing on the maximums and minimums obtained from the quantitative variables.

### 2.7. Descriptive Statistics

The variables of the study sample were analyzed by the Shapiro–Wilk and Kolmogorov–Smirnov Test for normality, to determine the follow-up. The variables were found to follow a normal distribution and subsequently defined by mean, standard deviation, and interval for values. Cases and percentages defined the discrete variables.

### 2.8. Analytical Statistics

Different tests were performed, such as the “analysis of variance ANOVA or the repeated measures ANOVA” to determine if there were significant differences between the three measurements of the patients in the variables that measure cognitive impairment (before, during, and after chemotherapy). 

The data processing was performed using IBM SPSS 23.0.

## 3. Results

A total of 174 patients were interviewed. Two did not want to participate in the study, and one withdrew the informed consent.

A total of 171 patients performed the study. Twenty did not fully complete the self-administered tests and, although in all other measures it was valid, failure to answer some item in a given test nullified the entire study for those patients.

The study comprises a total of 151 patients, as shown in the flow chart, Figure 1.

The first objective studied, analyzed the impact of chemotherapy on the cognitive domains in the three measurements studied.

The analysis of those domains or cognitive functions was as follows—low scores indicate an increase in cognitive impairment, except in the TMT-A and TMT-B tests, where the high scores indicate a significant increase in cognitive impairment.

A general pattern was met in most cognitive domains studied—as chemotherapy progressed, the cognitive performance of the patients in the study worsened significantly from the obtained scores at baseline.

Regarding memory (measured by the Wechsler Vocabulary Scale Subtest), we observed a decrease during treatment (M = 26.53) with regards to the start (M = 27.76), and recovery two months after completing chemotherapy, almost reaching the starting data (M = 27.46)

In contrast, with regards to the evolution of the processing speed (measured by the Symbol Search Subtest (SS), Key Search Test (KS)), attention (measured by the Trail Making Test (TMT) and the Stroop Test (ST)), a decrease was observed throughout the chemotherapy (SS M = 23.41; KS M = 38.32; TMTA M = 53.95; TMTB M = 121.42; STpyc M = 43.75). The lowest cognitive performance score was obtained two months after the end of treatment (Table 2).

The second objective was to assess whether the emotional state affects cognitive performance throughout the treatment, and to determine if there were significant differences in the tests that measured cognitive impairment, according to the emotional state of the patients (hospital anxiety and depression scale—HAD). The Analysis of Variance Test for independent samples, ANOVA Inter, was performed to determine if the emotional state of the patients affected cognitive performance. 

In this case, only the variables with statistically significant differences between the groups are shown. 

Significant differences were found in the following tests that were used to measure the cognitive domains studied between women classified as normal (anxiety and depression) (1), borderline case (anxiety and depression) (2), and emotional state with a clinical problem (anxiety and depression) (3) (Table 3): -Symbol Search before (F = 4.234; *p* < 0.05);-Letters and Number before (F = 5.152; *p* < 0.01);-Stroop Word before (F = 4.746; *p* < 0.001);-Stroop Color and Word before (F = 7.582; *p* < 0.01) and-Stroop Color and Word during (F = 7.102; *p* < 0.01)

Since the Emotional Status Factor had three comparison groups, Scheffé’s Post-Hoc Test was performed to determine which groups presented differences in the tests and which cognitive domains were affected.

Therefore, after analyzing the data of the patients’ situation before treatment, the differences were more evident in those whose emotional state (anxiety and depression) was classified as a clinical problem. Additionally, a worsening of cognitive performance was observed in the normal and borderline case. 

These differences occurred before and during chemotherapy, after which the values tended to normalize, that is, the performance improved.

The last objective studied determined if there were changes in the quality of life and if this affected cognitive performance (Table 4).

The statistical analyses performed by the inter ANOVA test did not find significant differences between patients who obtained a bad, regular, or good quality of life in the tests; neither before, during, or after treatment in any of the tests evaluating the cognitive domains. 

In this case, the classification for each patient, according to the scores obtained in the quality of life assessment (EORTC QLQ BR23), could be a section that was too subjective to find significant differences between those who were classified in one or another category. Therefore, it was complicated to find differences between them when we measured cognitive functions and impairment.

## 4. Discussion

The main objective of our research was to determine cognitive impairment in a sample of locoregional breast cancer patients as a consequence of chemotherapy.

The results report worsening of cognitive performance in most of the studied domains during chemotherapy. Most studies confirmed that cognitive disorders occurred in 15–50% of patients [31,32].

The bibliographical review showed that some patients with chemotherapy present memory and concentration problems [33]. This symptomatology was usually brief in most cases and tended to disappear when chemotherapy ceased, but in some cases, the symptoms could have repercussions on their quality of life, since they continued to cause some difficulties performing their daily life activities [34,35].

During chemotherapy, the cognitive performance of the patients in the tests decreased regarding the pre-treatment measurement. The performance improved after the end of the treatment, that is, in most cases, the cognitive performance improved with regards to the previous evaluation of the patients. These results were consistent with literature findings in which high levels of cognitive impairment were observed in patients receiving adjuvant chemotherapy, when comparing their performance in the tests with normative data or in patients who did not require chemotherapy or healthy persons. This supports the hypothesis that systemic chemotherapy treatment produces cognitive deficits [36].

Chemotherapy-related cognitive impairment is not a global phenomenon, but rather is specific to certain cognitive domains. Different studies do not agree, but it seems that the most frequently deteriorated domains are—attention span, verbal memory, working memory, processing speed, and motor [37,38].

In contrast, the executive functions are preserved [39,40]. Furthermore, the impairment in these domains is mild; that is, there is no severe and disabling impairment [16].

The cognitive domains evaluated in the research were selected based on the literature reviewed, which showed the most affected domains during chemotherapy. After analyzing the results, data showed that all cognitive domains studied were affected during chemotherapy, although not all showed the same impairment. The domains that were most affected in this research were processing speed and working memory. This might be because attention and memory processes are neuroanatomical structures associated with adult hippocampus neurogenesis. The new neurons generated by the hippocampus are essential for memory and learning, and brain-derived neurotrophic factor is required for growth [41].

The optimal functioning of processing speed, which refers to the amount of information that can be processed per unit time, was linked to the diameter of the nerve pathways, the integrity of the myelin sheath, the degree of myelination, the number of ion channels, and the efficiency of the synapses. Some studies linked the use of some chemotherapy drugs used in the treatment of breast cancer to cause toxicity in the Central Nervous System, which could cause cell damage and DNA chain breaks, leading to cell death and consequent synapse loss [42].

In the other cognitive domains studied, a tendency to worsen during chemotherapy was observed, but the results in these domains were not significant.

Kesler [43], performed a functional MRI in 25 women with breast cancer treated with chemotherapy, to 19 women with breast cancer who did not receive chemotherapy, and 18 healthy women, to determine which part of the brain was activated when the women performed a card-sorting test. The outcome showed that women who were treated with chemotherapy had significantly reduced function in the prefrontal cortex, the area of the brain responsible for skills such as problem-solving, working memory, and multitasking. They also had more errors and lower processing speed. 

Additionally, reduced left caudal lateral prefrontal cortex activation significantly correlated with a higher disease severity [23].

The results of this research were similar to those obtained in our study, except for the correlation found between cognitive impairment and disease severity.

The emotional state affected cognitive performance, as it increased cognitive impairment of patients during chemotherapy. The patients who had severe levels of anxiety and depression were those who suffered the most from cognitive performance. 

Some patients in our research showed high levels of anxiety and depression before starting treatment with chemotherapy. This state might be due to the impact of the recent diagnosis on the assessment of their cognitive performance. High levels of anxiety and depression can affect structures related and connected to the hippocampus, which could result in a worsening of patients’ cognitive performance.

These results are consistent with the literature reviewed, which states that anxiety and depression affect cognitive function, especially attention, and therefore the way the brain processes and stores information [44].

The study conducted by Jalali [45] showed a significant relationship between memory complaints and anxiety–depressive symptoms.

The impact of cognitive performance on the quality of life of oncological patients in our study showed that cognitive functioning had no impact. As some studies suggest [16,46], for now, the EORTC scales, in this case, QLQ-BR23, are not very useful to detect associated neuropsychological impairment. However, they are beneficial to evaluate the quality of life of oncological patients, according to the type of carcinoma and the associated symptoms.

## 5. Conclusions

Based on the results obtained we can say that the cognitive performance of breast cancer patients decreased throughout chemotherapy treatment. Recovery from this performance is not achieved in its entirety within two months after the end of chemotherapy treatment, which is an important factor in the worsening of cognitive performance and leading to symptomatology of depression or anxiety.

## Figures and Tables

**Figure 1 ijerph-17-08896-f001:**
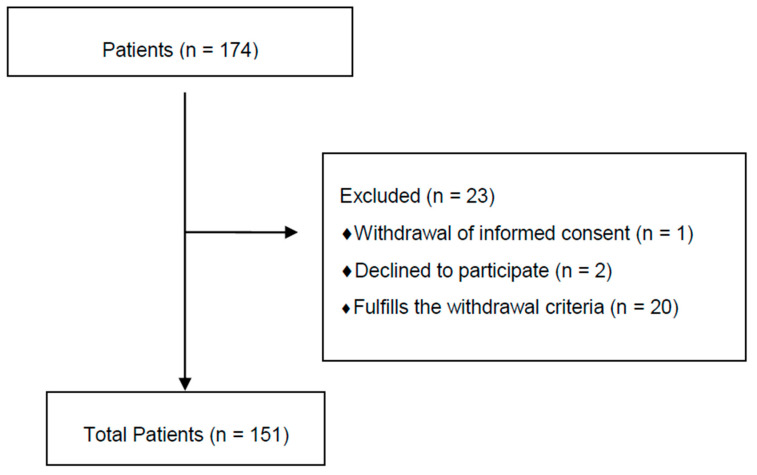
Flow chart of participants.

**Table 1 ijerph-17-08896-t001:** Exclusion and withdrawal criteria.

Exclusion Criteria	Withdrawal Criteria
Having locoregional breast cancer stage IIIA or above	Nonresponse of all the items in the questionnaires
Previous chemotherapy	Psychopharmacological treatment
No current chemotherapy treatment	Not completing the study follow-up
No patient of Salamanca University Assistance Complex	Exitus
Being a minor	
Age older than 85 years	
Pregnancy	
Not signing the written informed consent	
Chronic insomnia	
Psychopathological diagnosis	

**Table 2 ijerph-17-08896-t002:** Evolution of the domains studied in the three measurements.

Variables	M	SD	Minimum Score Obtained	Maximum Score Obtained
Vocabulary Before	27.76	6.590	15	45
Vocabulary During	26.53	7.427	14	47
Vocabulary After	27.46	7.127	15	46
Symbol search Before	25.71	5.650	15	41
Symbol search During	23.41	5.110	14	37
Symbol search After	22.59	5.078	13	39
Key Before	44.15	10.708	28	81
Key During	38.32	9.524	25	74
Key After	35.84	9.156	23	77
Before	15.42	1.757	11	21
L and N During	13.58	1.741	9	19
L and N After	12.22	1.496	9	16
TMT_A Before	50.85	16.486	27	89
TMT_A During	53.95	16.845	29	91
TMT_A After	55.81	17.199	31	95
TMT_B Before	116.75	32.853	65	198
TMT_B During	121.42	33.086	66	69
TMT_B After	125.23	33.25	69	209
Stroop word Before	122.27	4.286	110	132
Stroop word During	117.77	3.921	108	127
Stroop word After	114.93	3.871	107	126
Stroop color Before	76.86	4.032	70	86
Stroop color During	70.62	4.415	60	81
Stroop color After	67.56	3.834	59	79
Stroop color and word Before	49.30	4.064	40	59
Stroop color and word During	43.75	3.912	33	53
Stroop color and word After	41.94	3.077	31	49

M = Mean; SD = Standard deviation; Before—first evaluation under study, beginning of chemotherapy; during—second evaluation under study, three months after receiving treatment; after—last evaluation under study, two months after treatment ended.

**Table 3 ijerph-17-08896-t003:** Influence of anxiety and depression before treatment in the tests used.

Variables	HAD Emotional State: Anxiety and Depression					
	1. Normal *n* = 6	2. Borderline *n* = 62	3. Clinical Problem *n* = 83	Scheffé	F	*p*
	M	M	M			
Search Symbol before	26.00	27.24	24.54	2–3 *	4.234	*p* < 0.05
L y N before	17.00	15.73	15.08	1–3 *	5.152	*p* < 0.01
TMT_A before	38.83	44.24	56.66	1–3 * y 2–3 *	13.725	*p* < 0.001
TMT_A during	40.83	47.45	59.76	1–3 * y 2–3 *	13.219	*p* < 0.001
TMT_A after	42.67	49.03	61.82	1–3 * y 2–3 *	13.585	*p* < 0.001
TMT_B before	88.83	103.77	128.46	1–3 * y 2–3 *	14.478	*p* < 0.001
TMT_B during	93.33	107.97	133.49	1–3 * y 2–3 *	15.247	*p* < 0.001
TMT_B after	97.67	111.29	137.64	1–3 * y 2–3 *	15.936	*p* < 0.001
Stroop word before	123.83	123.39	121.33	2–3 *	4.746	*p* < 0.001
Stroop Color and Word before	50.83	50.65	48.18	2–3 *	7.582	*p* < 0.01
Stroop Color and Word during	44.17	45.10	42.72	2–3 *	7.102	*p* < 0.01

M = Medium of the variables. * Significant differences between the two groups in Scheffé’s test. Before—first evaluation under study, beginning of chemotherapy; during—second evaluation under study, three months after receiving treatment; after—last evaluation under study, two months after treatment ended; and TMT—Trail Making Test.

**Table 4 ijerph-17-08896-t004:** Development of Quality of Life throughout treatment*.

Quality of Life. Scores of the EORTC QLQ-BR23 Scale								
BEFORE			DURING			AFTER		
Low	Half	High	Low	Half	High	Low	Half	High
0 (0%)	19 (12.65%)	1 (7%)	15 (9.9%)	5 (3.3%)	0 (0%)	15 (9.9%)	5 (3.3%)	0 (0%)
4 (2.6%)	49 (32.5%)	12 (7.9%)	9 (6%)	55 (36.4%)	1 (7%)	1 (7%)	62 (41.1%)	2 (1.3)
2 (1.3%)	46 (30.5%)	18 (11.9%)	4 (2.6%)	44 (29.1%)	18 (11.9%)	4 (2.6%)	33 (21.9%)	29 (19.2%)
1 (0.7%)	20 (13.2%)	3 (2%)	16 (10.6%)	8 (5.3%)	0 (0%)	15 (9.9%)	9 (6%)	0 (0%)
5 (3.3%)	77 (51%)	18 (11.9%)	12 (7.9%)	80 (53%)	8 (5.3%)	5 (3.3%)	78 (51.7%)	17 (11.3%)
0 (0%)	17 (11.3%)	10 (6.6%)	0 (0%)	16 (10.6%)	11 (7.3%)	0 (0%)	13 (8.6%)	14 (9.3%)
1 (7%)	21 (13.9%)	3 (2%)	17 (11.3%)	8 (5.3%)	0 (0%)	15 (9.9%)	10 (6.6%)	0 (0%)
2 (1.3%)	35 (23.2%)	9 (6%)	6 (4%)	32 (21.2%)	8 (5.3%)	3 (2%)	30 (19.9%)	13 (8.6%)
3 (2%)	58 (38.4%)	19 (12.6%)	5 (3.3%)	64 (42.4%)	11 (7.3%)	2 (1.3%)	60 (39.7%)	31 (20.5%)

Study of the quality of life related to health in the three moments studied by the participants. Before—first evaluation under study, beginning of chemotherapy; during—second evaluation under study, three months after receiving treatment; after—last evaluation under study, two months after treatment ended.

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
