# Peer review of "Study of Chemotherapy-Induced Cognitive Impairment in Women with Breast Cancer"

_ijerph, 2020, doi:10.3390/ijerph17238896_

Round 1
Reviewer 1 Report
The authors of the Manuscript "Study of Chemotherapy-Induced Cognitive Impairment in Women with Breast Cancer" address the very important not sufficiently investigated phenomena of cognitive impairment after chemotherapy in patients with early local breast cancer. As the well written, thoroughly reviewed introduction very comprehensively describes, the transient and in some cases chronic impact of chemotherapy very often outperforms the positive impact of the chemotherapy on the prognosis of these cancers with relatively good prognosis. I was very much looking forward to reading the study and was disappointed by the data presented and the analysis performed.
The study needs major revision
- The methods are not thoroughly described, the tests chosen are not referenced, a brief description would help to understand which parameters were investigated. No justification was given for why these analyses were chosen over others and which alternatives were considered.
- Chronic fatigue is one of the main side effects accompanying the described cognitive impairments known as 'chemo-brain' I would have wished to have learned a little bit more about this very important side effect that surely impacts cognitive function and how the two are linked to each other
- The side effects should be correlated severity to treatment outcome and quality of life. I would have wished to see a correlation and appropriate stats to investigate this. Does the good outweigh the evil?
- Table 1 needs statistics. Visual illustration such as boxplots would markedly improve the paper. This was actually announced (lane 221) but not delivered by the authors
- The purpose, make and structure of table 3 is entirely unclear to me, please provide further labeling
- No details are given about the type of chemotherapy and treatment regiment
- No justification for the choice of statistical methods are provided
- If programs were used, the make and provider should be stated (example: SPSS 23.0, lane 232)
- Many appriviations and test methods are not explained (e.g. CAUSA (lane 167), TMT_A (lane 283), Stroop (lane 283), ADH (lane 283) is not
- Figure 1 is actually a table
- The study group is too small. If the methods chosen to describe the outcome would have been statistically accurate this would have been a very interesting pilot study to raise awareness leading to funding of a more comprehensive investigation
- The author should provide information about the country their institution is located in (lanes 6-10)
Author Response
REVIEW 1
Reviewer comment: The authors of the Manuscript "Study of Chemotherapy-Induced Cognitive Impairment in Women with Breast Cancer" address the very important not sufficiently investigated phenomena of cognitive impairment after chemotherapy in patients with early local breast cancer. As the well written, thoroughly reviewed introduction very comprehensively describes, the transient and in some cases chronic impact of chemotherapy very often outperforms the positive impact of the chemotherapy on the prognosis of these cancers with relatively good prognosis. I was very much looking forward to reading the study and was disappointed by the data presented and the analysis performed.
Research team response: We appreciate the comments of the reviewers and proceed to make the appropriate modifications.
The study needs major revision
- Reviewer comment: The methods are not thoroughly described, the tests chosen are not referenced, a brief description would help to understand which parameters were investigated. No justification was given for why these analyses were chosen over others and which alternatives were considered.
Research team response: We fully agree with the reviewer. We add citations and specify why we chose these evaluation tests for the variables under study.
- Reviewer comment: Chronic fatigue is one of the main side effects accompanying the described cognitive impairments known as 'chemo-brain' I would have wished to have learned a little bit more about this very important side effect that surely impacts cognitive function and how the two are linked to each other
Research team response: We agree with the comment raised by the reviewer. We note the suggestion for future studies, since it may be a good reason for study and correlation between symptomatic variables.
- Reviewer comment: The side effects should be correlated severity to treatment outcome and quality of life. I would have wished to see a correlation and appropriate stats to investigate this. Does the good outweigh the evil?
Research team response: We appreciate the considerations and specify. We consider the exposed statistical study appropriate, it was also the procedure proposed by an external statistical team specialized in clinical research.
- Reviewer comment: Table 1 needs statistics. Visual illustration such as boxplots would markedly improve the paper. This was actually announced (lane 221) but not delivered by the authors
Research team response: In full agreement with the reviewer, we make the appropriate modifications.
- Reviewer comment: The purpose, make and structure of table 3 is entirely unclear to me, please provide further labeling
Research team response: We agree. We proceed to further specify Table 3 (After the modifications it has become Table 4)..
- Reviewer comment: No details are given about the type of chemotherapy and treatment regiment
Research team response: We consider it appropriate not to provide details on the type of chemotherapy, since we think that we must first study whether or not there is chemoinduced cognitive impairment, to later analyze which type of chemotherapy this impairment comes from to a greater extent. We believe that it is necessary to go step by step in the study of chemoinduced deterioration, but we take it as a consideration for future studies.
- Reviewer comment: No justification for the choice of statistical methods are provided
Research team response: We agree, we proceed to make a more extensive explanation in the section "statistical methods"
- Reviewer comment: If programs were used, the make and provider should be stated (example: SPSS 23.0, lane 232)
Research team response: We agree with the reviewer's suggestion. We proceed to make the change.
- Reviewer comment: Many appriviations and test methods are not explained (e.g. CAUSA (lane 167), TMT_A (lane 283), Stroop (lane 283), ADH (lane 283) is not
Research team response: We modify according to what was raised by the reviewer. In the tables we continue to keep the abbreviations since we consider it would be overloading the reader, but we make sure that we have previously specified the meaning of the abbreviation.
- Reviewer comment: Figure 1 is actually a table
Research team response: We fully agree with the reviewer, we made the proposed modification.
- Reviewer comment: The study group is too small. If the methods chosen to describe the outcome would have been statistically accurate this would have been a very interesting pilot study to raise awareness leading to funding of a more comprehensive investigation
Research team response: We understand the consideration provided by the reviewer and propose future larger studies on the study of chemoinduced cognitive impairment.
- Reviewer comment: The author should provide information about the country their institution is located in (lanes 6-10)
Research team response: We fully agree with the reviewer. We make timely modifications.
Reviewer 2 Report
The authors here have reported a study of chemotherapy-induced cognitive impairment in women with stage I, II and III locoregional breast cancers who received chemotherapy at the University hospital of Salamanca during the course of the study. Partial cognitive impairment due to chemotherapy for cancer treatment is a well-known fact that is supported by ample clinical data. However, new studies in specific chemotherapeutic treatments in different demographics are always needed to obtain updated database with all possible variabilities. Under this context, the current work is important. The authors have put an extensive background of the work with well described data and discussion. Hence, I recommend the manuscript to be published in its current form.
Author Response
REVIEW 2
Reviewer comment: The authors here have reported a study of chemotherapy-induced cognitive impairment in women with stage I, II and III locoregional breast cancers who received chemotherapy at the University hospital of Salamanca during the course of the study. Partial cognitive impairment due to chemotherapy for cancer treatment is a well-known fact that is supported by ample clinical data. However, new studies in specific chemotherapeutic treatments in different demographics are always needed to obtain updated database with all possible variabilities. Under this context, the current work is important. The authors have put an extensive background of the work with well described data and discussion. Hence, I recommend the manuscript to be published in its current form.
Research team response: We greatly appreciate the comments provided by the reviewer. Thank you very much.
Reviewer 3 Report
The authors reported that chemotherapy induces cognitive impairment in breast cancer patients.
Major points.
- The overall sentences should be clearly and concisely organized.
Introduction (lines 66-150)
Materials and methods (lines 152-232)
- It is needed to re-write the tables and figures that they can be understood by themselves.
Figure 1 (This might be table form)
Table 1
Table 2
Table 3
Minor points
- Define abbreviations at first mention.
- line 194
Cognitive functions?
- lines 200-202
Add the references for the tests
Author Response
REVIEW 3
Major points.
- Reviewer comment: The overall sentences should be clearly and concisely organized.
Introduction (lines 66-150)
Materials and methods (lines 152-232)
Research team response: Thank you very much, we have already adapted the article with your recommendation.
- Reviewer comment: It is needed to re-write the tables and figures that they can be understood by themselves.
Figure 1 (This might be table form)
Table 1
Table 2
Table 3
Research team response: We have modified the tables to make them more understandable.
Minor points
- Reviewer comment: Define abbreviations at first mention.
- Reviewer comment: line 194 Cognitive functions?
Research team response: Yes, they are cognitive functions, we have already added it in the text to clarify it.
- Reviewer comment: lines 200-202. Add the references for the tests
Research team response: Thank you very much for the clarification. We have already added the references in the text and in the bibliographic references section.
Round 2
Reviewer 3 Report
I checked the response of the authors.